# Blood RNA biomarkers and a point-of-care elastase assay for detecting host immune activation in suspected sepsis: Trajectory matters

John E. Lafleur[1], Eduard Shaykhinurov[2], John Perkins[3], Richard Wargowsky[3], Kevin Jaatinen[3], Mary Pasquale [3], Grace Holloway [3], David Yamane[1,2], Akhil Patel[2], Daniel King[2], Andrew Meltzer[1], Ryan Heidish[1], Soroush Shahamatdar[1], Aditya Loganathan[1], Tarun Loganathan[1], Taylor Bolden[1], Michael Zane Hayden[3], Aditya Maddali[4], Jennifer Goldman [3], Zachary Falk[3], Tisha Jepson[3], Avery League[3], Timothy A. McCaffrey [3]*

1 Department of Emergency Medicine, The George Washington University Medical Center, Washington, DC, 2 Department of Anesthesiology and Critical Care Medicine, The George Washington University Medical Center, Washington, DC, 3 Department of Medicine, Division of Genomic Medicine, The George Washington University Medical Center, Washington, DC, 4 Department of Medicine, The George Washington University Medical Center, Washington, District of Columbia

* mcc@gwu.edu

## Abstract

### Introduction/Hypothesis

The chances of survival from sepsis are improved by early diagnosis and treatment. In the prospective SENSOR study, RNA biomarkers of innate immune neutrophil activation were examined in emergency department (ED) patients triggering an automated sepsis alert. We hypothesized that higher levels of blood RNA biomarkers related to neutrophil activation would be associated with progression to more severe forms of sepsis.

### Methods

Adult patients in the ED triggering a sepsis alert were consented, enrolled, and study samples were obtained during the ED visit. Additionally, study samples were collected from a convenience sample of 16 adult non-ED controls and 8 other adults with self-described infectious illnesses. Blood samples were drawn in RNA preservative (Tempus) and whole blood RNA was analyzed by droplet digital PCR (ddPCR) for RNA transcripts related to neutrophil response to infection by bacterial (DEFA1; ALPL, IL8RB/CXCR2), and viral (IFI27, RSAD2) pathogens. Bacterial burden in blood was quantitated by ddPCR of 16S ribosomal DNA. Separately, neutrophil elastase was measured in immunomagnetically captured CD66b+ neutrophils by a novel point-of-care device.

**Data availability statement:** All relevant data are within the manuscript and its Supporting Information files.

**Funding:** The Ulvi and Reykhan Kasimov Family The St. Laurent Institute NIH S10 OD021622 The funders had no role in study design, data collection and analysis, decision to publish, or preparation of the manuscript.

**Competing interests:** TM and TJ have an equity interest in True Bearing Diagnostics, Inc., a diagnostics company developing RNA biomarkers for various diseases, including internal infections. TM has both issued and pending patents for technology related to the current studies. This does not alter our adherence to Journal policies on sharing data and materials. The other authors declare there are no competing interests.

**Abbreviations:** CBC: complete blood count; ddPCR: droplet digital PCR; DEGs: differentially expressed genes; ED: emergency department; ELISA: enzyme-linked immunosorbent assay; NETs: neutrophil extracellular traps; RIN: RNA integrity number; RNAseq: RNA sequencing; rRNA: ribosomal RNA; SIRS: systemic inflammatory response syndrome; WBC: white blood cell.

## Results

Patients were grouped using both 'Sepsis-2' SIRS criteria, adjudicated by independent physicians, and 'Sepsis-3' criteria which uses a qSOFA score for categorizing the severity of illness. Across 72 enrolled sepsis alert patients, 62.5% showed positive RNA biomarkers for bacterial infection, and 8.3% were positive for viral markers, with only 2 cases that showed only a viral signal. Septic patients showed a 4-fold increase in RNA markers vs. those without infection ($p < 0.05$). However, no significant differences were observed in RNA levels between those with sepsis vs those with more severe forms of sepsis. Likewise, RNA biomarker levels did not discriminate patients with qSOFA≥2 from qSOFA = 1. In a subset of patients with zero and three hour blood samples (n = 27) it was found that changes in RNA levels (up or down), or neutrophil elastase activity, was strongly associated with progression to more severe forms of sepsis or a qSOFA score of ≥2.

## Conclusions

Patients progressing to more severe forms of sepsis did not have higher absolute levels of neutrophil activation RNA biomarkers (or higher bacterial burden in blood) compared to patients with sepsis. However, a change in RNA biomarkers or elastase between zero and three hours was strongly indicative of progression to more severe forms of sepsis.

## Introduction

Historically, sepsis had been defined by physiologic parameters, including systemic inflammatory response syndrome (SIRS) criteria [1], which lacked specificity, and while relatively sensitive, still failed to capture almost 13% of sepsis cases [2]. In 2016, sepsis was redefined as 'severe organ dysfunction resulting from immune dysregulation in response to infection' [3]. This updated definition, known as Sepsis-3, led to greater diagnostic specificity, but loss of sensitivity, due to a more restricted list of clinical and laboratory findings that qualify an infected patient as 'septic' [4,5]. Despite exhaustive research, improved means for screening for sepsis have not materialized. Recent algorithmic approaches based upon the electronic health record (EHR) have failed to improve diagnosis [6]. The Centers for Medicare & Medicaid Services (CMS) mandates the collection of quality metrics around sepsis care based on a pre-Sepsis-3 framework, and most medical systems continue to screen for sepsis using pre-2016 'SIRS' criteria [7], The Sepsis-3 definition relies on parameters included in a clinical predictive tool known as the 'Sequential Organ Failure Assessment' (SOFA), and 'SOFA' requires lab values which are usually not available in the time-frame required in the ED for early recognition of patients who may be septic. Thus, an alternative measure, quick SOFA (qSOFA), has been adopted. However, past research shows that the sensitivity of the qSOFA score during ED triage is as low as 31% for predicting admission to the Intensive Care Unit (ICU) and 60% for

predicting mortality at 48 hours [4]. 'Severe sepsis' which is important in 'Sepsis-2' (sepsis plus end organ dysfunction), was removed in 'Sepsis-3'. For purposes of convenience, because results in the present study were analyzed according to criteria from both 'Sepsis-2' and 'Sepsis-3', in the following the term 'more severe form of sepsis' has been used to denote those with 'severe sepsis/septic shock' under 'Sepsis-2', and those with a qSOFA score ≥2 in 'Sepsis-3'.

There is a well-recognized need for novel biomarkers and a better understanding of the dynamics of the transition to more severe forms of sepsis [8]. There has been an extensive search for sepsis biomarkers, with possibly ~250 identified over the past few decades, but few have proven better than a functional measure of organ failure, such as the SOFA score [9–11], which is largely an ICU-based measure not suitable for application to emergency department patients where most sepsis is diagnosed.

Adherence to the tenets of 'early goal-directed therapy' [12], had been considered beneficial, however its value compared to 'usual care' is now questioned [13]. About 20% of deaths worldwide [13] are due to sepsis, and about 20% of patients presenting to the ED trigger sepsis alerts based upon current screening practices—though many turn out to have no infection [6]. Aggressively treating all ED patients who trigger sepsis alerts is a costly, time-intensive means for improving survival. Thus, improved diagnostics are needed to identify the patients that require aggressive treatment for sepsis.

A large and growing body of literature implicates excessive neutrophil activation and cell death in the progression of sepsis (reviewed in [14]). In previous work, we employed whole transcriptome profiling to identify RNA biomarkers in peripheral blood from patients with appendicitis or respiratory infections [15], and in various types of COVID-19 patients [16]. Relatively large fold-change increases (~20-fold) were identified in RNAs coding for neutrophil primary granule markers, especially neutrophil α-defensins (DEFA1) in infections such as pneumonia. In appendicitis patients, there was marked increases in RNAs coding for IL8 receptor-ß (IL8RB/CXCR2), and secondary granule proteins such as alkaline phosphatase (ALPL) [15]. Neutrophil RNA markers of viral infection, IFI27 and RSAD2, were identified from COVID19 cases and the literature to provide a more complete picture of the infection profile [16–18]. Recent work suggests the trajectory of sepsis biomarkers over time can provide more diagnostic value than measurements of these markers at a single time point [9,19]. In a subset of patients in the current study samples were drawn at time zero (patient enrollment in the study), and again three hours later. Additionally, to further understand the interactions between pathogens and host immunity, bacterial 16S and fungal 23S DNA were assayed, as were DEFA1 *protein* levels.

Here we report on a set of promising RNA biomarkers in whole blood, and elastase activity of circulating CD66b neutrophils, to evaluate whether they can help identify ED patients at risk for developing more severe forms of sepsis early in the ED course, when resuscitative measures are most useful for preventing the worst outcomes.

## Materials and methods

### Enrollment and blood draw

This prospective, observational study, shown schematically in Fig 1, was approved by the Institutional Review Board of The George Washington University (#NCR213645). All subjects, or their legally authorized surrogate, provided written informed consent, witnessed by a study team member in the period from 11/16/2022 through 8/21/2024. Patients presenting to The George Washington University Hospital ED were screened for inclusion criteria. Inclusion criteria consisted of a 'sepsis alert' trigger generated automatically by the Cerner EHR using modified SIRS criteria shown in Fig 1 [20]. The Cerner sepsis alert uses The St. John's Sepsis alert that is similar to SIRS criteria: temperature >38.3°C or <36°C, heart rate > 95 bpm, respiratory rate ≥22 breaths/min, white blood cell count >12 or <4 K/ul, or glucose >141 and <200 mg/dl. The alert registers when 3 criteria are met, or 2 criteria and evidence of organ failure: hypotension, elevated lactate, elevated bilirubin or creatinine.

## SENSOR Study Design

| Emergency Department Visit | Analysis 1<br>Sepsis-2 Criteria | Analysis 2<br>Sepsis-3 Criteria | Molecular Testing |
|---|---|---|---|
| **Enrollment Criteria:**<br>• Fever<br>• Tachycardia<br>• Elevated Respiration<br>• High/Low white count<br>• Abnormal glucose | **Sepsis/Shock**<br>Infection with End organ dysfunction or Hypotension | **qSOFA ≥ 2**<br>Altered mental status, hypotension, elevated respiratory rate | **Host response to infection:**<br>**RNA Markers (ddPCR)**<br>• Bacterial1=DEFA1<br>• Bacterial2= ALPL+IL8RB<br>• Viral= IFI27+RSAD2 |
| | **Sepsis**<br>Infection w/o organ dysfunction | **qSOFA = 1**<br>Only one of above criteria | **Cytocapture of Neutrophils**<br>• Kinetic elastase assay<br>• Point of Care |
| **ED Evaluation:**<br>• Automated sepsis alert<br>• Imaging<br>• Labs/Microbiology | **NO Infection**<br>Post-hoc adjudication | **qSOFA = 0**<br>None of above criteria | **Trajectory of markers**<br>• 0 hr to 3 hr change |
| **NON-ED Subject Recruitment** | Infected Non-ED | | |
| | Normal Controls | | |

**Fig 1. Design of the SENSOR study.** Patients presenting to the emergency department (ED) were screened manually and by an electronic health record algorithm to detect a risk of sepsis using fever, heart rate, respiration, white blood cell (WBC) count, and other criteria. Patients meeting criteria were consented, blood drawn into an RNA stabilizer for quantitation of RNA biomarkers of neutrophil activation using droplet digital PCR for the bacterial response markers RNA: DEFA1, ALPL and IL8RB/CXCR2, and the viral response marker RNAs: IFI27 and RSAD2. Blood was also collected in EDTA anticoagulant for determination of neutrophil elastase using a novel immunomagnetic isolation and kinetic elastase assay (CyBIS). The clinical course was not altered, but all relevant clinical results were recorded and used to adjudicate the diagnosis as involving No Infection, Sepsis, or more severe forms of sepsis (Sepsis/Shock).

Normal controls and subjects with self-reported infections were sampled from medical center personnel. After informed consent was obtained, a sample of whole blood was drawn into 2 Tempus Blood RNA tubes (ThermoFisher, 3 ml each), and 1 BD Vacutainer K2 EDTA tubes (BD Biosciences, 5 ml) for neutrophil capture and elastase assay. A subset of patients (n = 27) had a second blood sample drawn 3 hours after the initial sample in order to assess temporal changes in the biomarkers. Study personnel subsequently processing the blood samples were blinded to the subjects' clinical characteristics and outcomes by anonymous coding of the samples.

### Adjudication of infection and sepsis status

The presence of infection and sepsis was adjudicated from the medical records by two physicians unrelated to the ED care of the patients, and in the case of discordance was reviewed by a third physician. Determination of infection was determined by microbiologic testing (culture and non-culture based). In cases of negative microbiologic testing, clinical or radiographic diagnoses consistent with an infection were considered positive, for example perforated diverticulitis or pneumonia. More severe forms of sepsis were determined by an active systemic infection with signs of end organ dysfunction, including hypotension, acute kidney injury, altered mental status, or lactic acidosis. In a second analysis, classification was determined by calculating the qSOFA scores using respiratory rate ≥22, systolic blood pressure <100, and altered mentation (Glasgow Coma scale, GCS < 15), consistent with Sepsis-3 criteria [3]. The 'No Infection' subgroup was defined as patients that triggered the sepsis alert, but post-hoc analysis adjudicated that no infection was present by all available testing.

### Concordance of infection and sepsis adjudication

The inter-observer concordance for the presence of an infection was 87.5%, but the concordance for more severe forms of sepsis was only 69.4%, and concordance on both parameters was only 65.8%. While this seems low, it is consistent with other published data. In large clinical trials of other infection diagnostics, inter-observer concordance was found to be only 71.2% for both bacterial and viral infection adjudication [21].

### Neutrophil-associated elastase activity by cytocapture (CyBIS)

The method for automated cytocapture and neutrophil elastase assay is described previously [22]. Briefly, blood collected in EDTA anticoagulant (75 µl) is pelleted at 300 x g for 5 min at 4˚ C to remove soluble plasma ligands that can block the antibody beads. The pelleted cells are resuspended in bead binding buffer, containing EDTA, and are mixed with CD66b antibody-coated magnetic beads and then loaded into the CyBIS device. The device incubates the blood and bead mix for 20 minutes, collects magnetic beads with cells, and washes away unbound cells and proteins 3 times. The captured cells are lysed and assayed by time-dependent cleavage of a colorimetric elastase substrate, yielding a rate constant (Vmax, mOD/min) that is proportional to elastase levels in the captured cells. Cell counts were performed using 10 µl in a hemocytometer at 200X using phase contrast. The identities of the isolated cells were confirmed by fixing with 4% formaldehyde in 1X PBS, and using a May-Grünwald stain (Wright-Giemsa stain with eosin Y).

### Whole blood RNA extraction

The Tempus Blood RNA preservation tubes (Applied Biosystems) were stored at −80˚ C and thawed for 1 hour at RT. RNA was purified using the Tempus Spin RNA Isolation Reagents (Applied Biosystems). The Tempus blood lysate was spun at 3000 x g for 30 minutes, and the pellet was resuspended and transferred onto purification columns. Following on-column DNAse treatment using TurboDNase (Ambion) and a series of washing and drying steps, purified RNA was eluted with nuclease-free water. RNA levels were quantified using the NanoDrop ND-1000 Spectrophotometer and quality was determined by capillary electrophoresis on the Agilent 2100 Bioanalyzer.

### RNA Biomarker Digital Droplet PCR (ddPCR)

Prior microarray and RNAseq analysis of whole blood RNA from patients with various types of infections identified RNA markers of viral and bacterial infections [15,16,23]. Primers and internal fluorescent probes for the RNAs known to be responsive to bacterial infections (DEFA1, IL8RB/CXCR2 and ALPL) and to viral infections (IFI27 and RSAD2), and to ACTB as a reference control can be found in Supplementary Data S1 Fig. The DNAse-treated RNA from whole blood (200 ng) was reverse transcribed into cDNA using gene-specific primers (250 nM) with the RNAse H+ reverse transcriptase and 5X Supermix from the iScript Select cDNA Synthesis Kit (Bio Rad). Each of the markers were paired to make a total of three primer-probe master mixes (4 µM forward primer; 4 µM reverse primer, 1.12 µM quenched fluorescent probe) with water and the ddPCR Supermix (Bio Rad).

The BioRad Automated Droplet Generator typically produced >15,000 nanoliter-sized droplets per well, the target transcripts were amplified in each droplet by PCR, and each droplet was read separately by the BioRad QX200 Droplet Reader. BioRad Quantasoft software was used to set the threshold for positive droplets. Transcript counts for the RNA biomarkers (DEFA1, ALPL, etc.) were normalized per sample by dividing by the number of ACTB copies to further minimize the effects of changes in cell counts or in RNA yield.

### Defensin 1 protein in plasma

A subset of patients and controls had plasma from EDTA blood stored at −80˚C. Thawed plasma was analyzed for levels of the defensin-α1 protein by ELISA (RayBio ELH-DEFA1). Plasma was diluted 1:1 with assay buffer and reacted in the

sandwich ELISA assay and compared to a standard curve of defensin-α1 protein (0–250 ng/ml). The observed values were corrected for dilution and reported in ng/ml.

## 16S Bacterial and 28S Fungal ribosomal DNA quantitation

DNA was isolated from the Tempus tubes via a modified phenol/chloroform extraction procedure [24]. First, total lysis of any remaining viable cells was ensured by the addition of an 8% SDS solution to the Tempus blood/preservative mixture and multiple cycles of freezing in a dry/ethanol bath immediately thawing at 95˚C [25]. Two rounds of extraction with a phenol/chloroform/isoamyl mixture (25:24:1 ThermoFisher) and a final cleanup with a chloroform/isoamyl (24:1) mixture were then performed. The DNA was then precipitated overnight at −80˚C in the presence of 3M sodium acetate and glyco-gen (8 µg/µl). 16S (Gram positive/negative) and 28S RRNA fragments were then quantified by ddPCR using primers with minimized human cross-reactivity and high sensitivity and specificity in patients with proven central line infections [26].

## Statistical methods

Continuous variables, such as gene expression levels, were tested for normality using the Jarque-Bera test. Parametic tests, such as the Student's t-test, were used for continuous variables, while non-parametric tests, such as chi-square, were used for categorial variables such as sex. Fisher's exact test was used when categorical variables contained a group of less than 5. Error bars are shown using the standard error of the mean (S.E.M.). Intercorrelations of measures was computed using Pearson's R.

## Results

### Clinical characteristics of sepsis patients and controls

A total of 73 ED subjects were enrolled, but 1 subject failed to yield sufficient RNA for analysis, leaving 72 ED subjects that divided about evenly between subjects adjudicated with more severe forms of sepsis (n = 22), sepsis (n = 26), or judged to have no infection (n = 24). The clinical, demographic, and laboratory findings are shown in Table 1, and reflect the type of changes known for sepsis and infection including age, white blood cell count, neutrophil count, and lactate levels. While trends in these were identified, differences between the groups were mostly not statistically different. Aver-age lactic acid levels were higher than expected in the No Infection group (3.13 mmol/L) because of 2 patients with very high values (11–13 mmol/L) due to seizures or alcohol withdrawal. Normal control subjects, and non-ED subjects with self-reported infections, were more likely to be younger and white, without significant differences in gender or ethnicity (S1 Table).

### Blood RNA biomarkers and neutrophil elastase as a function of infection, sepsis, and more severe forms of sepsis

**RNA Biomarkers.** The blood RNA biomarkers were quantitated in each patient and then analyzed as a function of their clinically adjudicated status. Additionally, normal control subjects (n = 16) and non-ED volunteers with self-described infections (n = 8) were similarly assayed for neutrophil elastase activity and blood RNA biomarkers. As shown in Fig 2, the RNA biomarkers showed stepwise increases from normal healthy controls to normal controls with self-reported infectious illness to the ED patients. However, there were not significant differences in either the RNA or elastase measures between patients that developed severe sepsis versus those with sepsis and an adjudicated infection. In fact, DEFA1 RNA levels were similar between more severe forms of sepsis (Sepsis/Shock), sepsis, and non-ED patient controls with self-reported infections. These results were similar in the second analysis, where there were no significant differences in either the RNA or elastase measures across the qSOFA groups. While the average levels of RNA biomarkers were not statistically different, a higher percentage of patients with more severe forms of sepsis were positive for any one RNA biomarker using

**Table 1. Demographic and lab values.**

| | SEPSIS/ SHOCK | SEM | | SEPSIS | SEM | | NO INFECTION | SEM |
|---|---|---|---|---|---|---|---|---|
| **N** | 22 | – | | 26 | – | | 24 | – |
| **Age (yrs)** | 57.95 | 3.73 | * | 47.19 | 3.93 | | 48.71 | 4.38 |
| **Weight (kg)** | 88.28 | 5.73 | | 84.56 | 6.77 | | 94.91 | 6.34 |
| **Sex (% male)†** | 36.40 | – | | 50.00 | – | | 37.50 | – |
| **Race (% white)†** | 27.27 | – | | 38.46 | – | | 29.17 | – |
| **Body Temperature (˚F)** | 99.73 | 0.50 | * | 98.85 | 0.37 | | 98.40 | 0.35 |
| **Systolic Blood Pressure (mmHg)** | 129.77 | 9.00 | | 131.88 | 4.95 | * | 146.52 | 5.72 |
| **Diastolic Blood Pressure (mmHg)** | 75.86 | 4.39 | | 77.40 | 3.16 | | 85.91 | 4.78 |
| **Heart Rate (beats per minute)** | 118.45 | 5.84 | | 105.32 | 4.11 | | 114.52 | 4.01 |
| **Respiratory Rate (breaths per minute)** | 23.76 | 1.85 | | 20.05 | 1.13 | | 22.05 | 1.70 |
| **White Blood Cells (K/µL)** | 14.87 | 1.38 | | 13.46 | 1.11 | | 12.36 | 1.91 |
| **Red Blood Cells (M/µL)** | 4.13 | 0.23 | | 4.04 | 0.25 | | 3.93 | 0.18 |
| **Hemoglobin (g/dL)** | 12.24 | 0.72 | | 11.73 | 0.67 | | 11.21 | 0.56 |
| **Hematocrit (% packed RBC)** | 37.50 | 1.98 | | 36.05 | 2.00 | | 35.26 | 1.59 |
| **Mean Corpuscular Volume (fL)** | 91.40 | 1.87 | | 90.18 | 1.57 | | 90.51 | 2.35 |
| **Mean Corpuscular Hemoglobin (pg)** | 29.60 | 0.66 | | 29.27 | 0.59 | | 28.77 | 0.98 |
| **MCH Concentration (g/dL)** | 32.41 | 0.37 | | 32.46 | 0.39 | | 31.68 | 0.49 |
| **Red Cell Distribution Width (mean %)** | 15.37 | 0.68 | | 14.53 | 0.46 | | 15.47 | 0.59 |
| **Platelet Count (K/µL)** | 260.52 | 38.89 | | 327.62 | 42.27 | * | 222.21 | 35.96 |
| **Bands (%)** | 5.75 | 2.37 | | 0.27 | 0.20 | | 0.33 | 0.22 |
| **Neutrophils (%)** | 76.80 | 2.79 | * | 78.96 | 1.65 | * | 64.43 | 4.37 |
| **Lymphocytes (%)** | 13.50 | 2.46 | | 11.46 | 1.19 | * | 22.13 | 3.45 |
| **Monocytes (%)** | 7.47 | 0.80 | | 7.67 | 0.56 | | 8.61 | 1.06 |
| **Eosinophils (%)** | 0.38 | 0.15 | * | 0.88 | 0.33 | | 1.73 | 0.53 |
| **Basophils (%)** | 0.07 | 0.06 | | 0.08 | 0.06 | * | 0.45 | 0.18 |
| **Immature Granulocytes (%)** | 0.85 | 0.13 | * | 0.62 | 0.13 | | 0.40 | 0.03 |
| **Urine Leukocyte Esterase (% +)‡** | 50.00 | – | * | 25.00 | – | | 7.69 | 0.19 |
| **Urine Nitrate (% +)‡** | 22.22 | – | | 16.67 | – | | 0.00 | – |
| **Urine White Blood Cells (% +)†** | 76.19 | – | * | 41.67 | – | | 39.13 | – |
| **Urine White Blood Cell Count** | 191.06 | 95.31 | | 41.70 | 13.99 | | 1.78 | 0.33 |
| **Blood Culture (% +)‡** | 40.00 | – | * | 5.00 | – | | 0.00 | – |
| **C-reactive Protein (mg/L)** | 198.56 | 35.32 | | 146.76 | 22.51 | | 186.73 | 28.05 |
| **Lactic Acid (mmol/L)** | 3.59 | 0.51 | | 1.88 | 0.18 | | 3.13 | 0.82 |

Notes: Values are means unless otherwise indicated. Percent values calculated using the total number of patients with available data for each variable as the denominator. Asterisks indicate statistically significant differences between the sepsis/shock vs. sepsis and/or no infection groups and the sepsis vs. no infection groups based on p-values < 0.05 using equal variance two-tailed t-tests for continuous variables and either chi-squared tests or Fisher's Exact Tests for categorical variables based on sample size. SEM = Standard Error of the Mean. Parameters for Controls are shown in S1 Table.

† Chi-squared test

‡ Fisher's Exact Test (note: no infection group excluded in the blood culture and urine nitrate tests to avoid categories where N = 0)

preset thresholds: more severe sepsis (81.8%), sepsis (57.7%), and uninfected (50.0%). The patient-level, anonymous raw data is available in S2 Table.

**Correlation of RNA biomarkers and clinical laboratory values.** The correlation of RNA biomarkers and other clinical variables can be seen in Fig 3. It should be noted that RNA biomarkers are measured in such a way that variations in

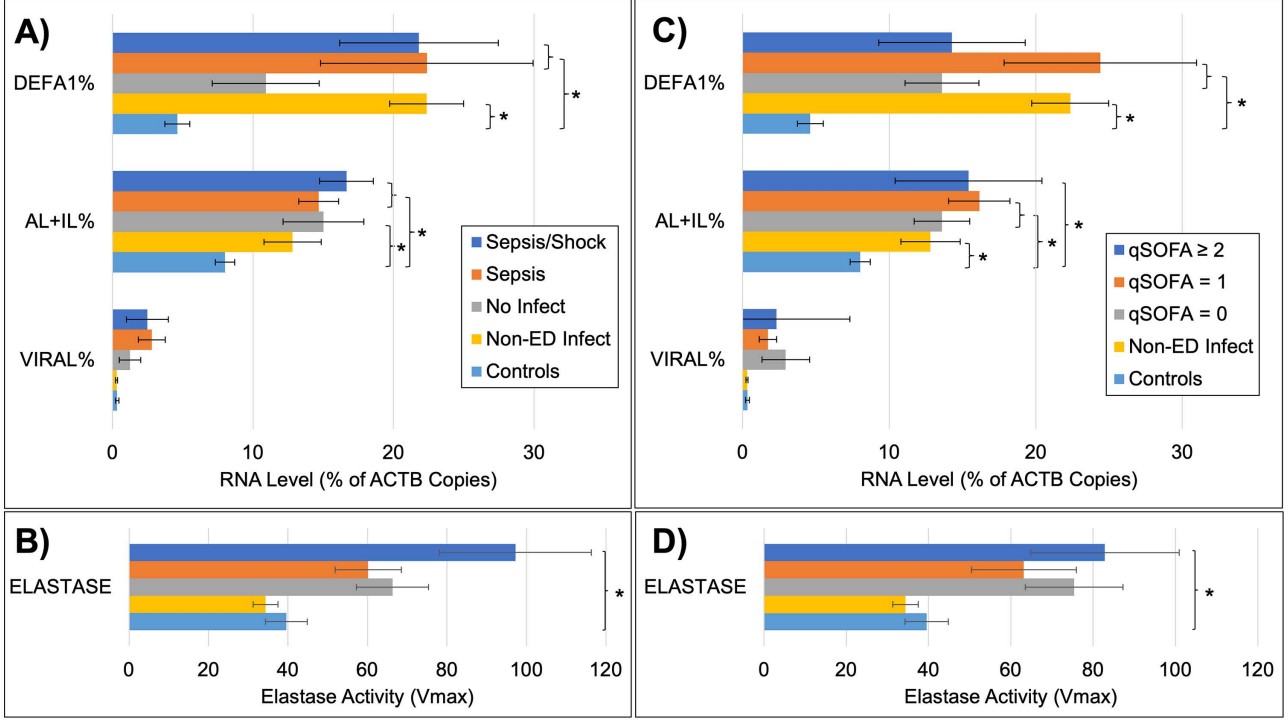

**Fig 2. RNA biomarkers and cytocaptured neutrophil elastase levels in controls and ED patients. Panel A:** RNA stabilized blood samples from Controls (n = 16), self-diagnosed but non-ED infections (Non-ED Infect n = 8), and ED patients adjudicated to have no infection (n = 24), sepsis (n = 26), or severe sepsis/septic shock (n = 22), were purified and quantitated by ddPCR for 6 host RNAs: ACTB1, a reference; DEFA1, ALPL and IL8RB/CXCR2 (AL + IL%), bacterial response markers; and IFI27 and RSAD2 (Viral%), viral response markers. The host response markers are expressed as a percent of ACTB reference (mean ± s.e.m., * = p < 0.05) **Panel B:** EDTA-treated blood was assayed for neutrophil elastase activity by immunomagnetic capture of neutrophils using CD66b-antibody coated beads followed by a kinetic assay for elastase that measures elastase activity over a 15 min period (Vmax, mOD/min). **Panel C**: The patients were regrouped according to their qSOFA scores and the RNA scores were re-averaged and compared between groups. **Panel D**: Cytocaptured neutrophil elastase activity is plotted according to the same qSOFA groupings as Panel **C.**

WBC count would have little impact on the RNA levels. The reasons for this are: 1) whole blood RNA contains significant RNA from RBC and platelets, thus minimizing the impact of WBC variation on total RNA yield, 2) regardless of RNA yield from the 3 ml of blood, a fixed amount of RNA is taken for ddPCR, and 3) the RNA biomarkers are normalized per ß-actin (ACTB) copies in the sample.

**Dynamic levels of immune activation in patients progressing to more severe forms of sepsis.** To understand the temporal stability of the RNA and elastase biomarker signals, and whether temporal changes could be relevant to sepsis progression, a subset of patients had a second blood draw (n = 27). Blood samples were obtained at the time of enrollment (0 hr), and 3 hours later.

As shown in Fig 4, the more severe sepsis patients showed a noticeable pattern of changes in the DEFA1 RNA level across the 3 hour period, with 4 patients showing abnormal scores with >25% change, up or down, in expression level. Likewise, ALPL and IL8RB/CXCR2 were quite variable in the more severe sepsis group with 3 additional patients showing abnormal scores with >25% change. Thus, 7/11 more severe sepsis patients had abnormal and changing scores. The sepsis group and non-infected group each had only 1 DEFA1 and 1 ALPL and IL8RB/CXCR2 score that was abnormal and changing (chi-square p < 0.05 more severe sepsis vs sepsis+uninfected). These results were similar in the second analysis, where 83% (10/12) of patients with a qSOFA score ≥ 2 had a > 25% change in their DEFA1 or ALPL and

| | Vmax | Path | DEFA P | %DEFA | %AL+IL | %VIRAL | TruNav | AGE | Temp | SYS BP | HR | WBC | RBC | Neut% | Lymph% | Mono% | ImmGr% | UrineWC | Glucose | Lactic | CRP |
|---|---|---|---|---|---|---|---|---|---|---|---|---|---|---|---|---|---|---|---|---|---|
| **Vmax** | 1.00 | | | | | | | | | | | | | | | | | | | | |
| **Pathogen** | 0.13 | 1.00 | | | | | | | | | | | | | | | | | | | |
| **DEFA Prot 0** | -0.25 | -0.30 | 1.00 | | | | | | | | | | | | | | | | | | |
| **%DEFA** | 0.01 | 0.35 | -0.45 | 1.00 | | | | | | | | | | | | | | | | | |
| **%BIOFILM** | 0.32 | 0.19 | 0.57 | 0.01 | 1.00 | | | | | | | | | | | | | | | | |
| **%VIRAL** | -0.15 | -0.02 | 0.17 | 0.03 | -0.09 | 1.00 | | | | | | | | | | | | | | | |
| **TruNav** | 0.10 | 0.37 | -0.07 | 0.93 | 0.32 | 0.17 | 1.00 | | | | | | | | | | | | | | |
| **AGE** | 0.01 | -0.09 | 0.16 | -0.19 | 0.19 | 0.02 | -0.11 | 1.00 | | | | | | | | | | | | | |
| **Temp** | -0.01 | -0.01 | -0.13 | 0.00 | -0.13 | 0.15 | -0.01 | -0.22 | 1.00 | | | | | | | | | | | | |
| **SYS BP** | -0.12 | -0.32 | -0.12 | -0.07 | -0.03 | 0.02 | -0.07 | 0.01 | -0.03 | 1.00 | | | | | | | | | | | |
| **HR** | -0.16 | 0.17 | -0.07 | 0.16 | -0.04 | -0.07 | 0.13 | -0.17 | 0.41 | 0.22 | 1.00 | | | | | | | | | | |
| **WBC** | 0.27 | 0.14 | -0.04 | 0.00 | 0.27 | -0.25 | 0.05 | -0.09 | 0.03 | -0.03 | -0.16 | 1.00 | | | | | | | | | |
| **RBC** | 0.04 | -0.20 | -0.34 | -0.08 | 0.20 | -0.02 | -0.01 | 0.02 | -0.08 | 0.01 | 0.06 | -0.01 | 1.00 | | | | | | | | |
| **Neut%** | 0.27 | 0.27 | -0.17 | 0.07 | 0.39 | -0.21 | 0.16 | 0.12 | -0.07 | -0.10 | -0.08 | 0.49 | 0.11 | 1.00 | | | | | | | |
| **Lymph%** | -0.27 | -0.23 | 0.24 | -0.09 | -0.39 | 0.18 | -0.18 | -0.13 | 0.01 | 0.15 | 0.17 | -0.54 | -0.12 | -0.82 | 1.00 | | | | | | |
| **Mono%** | -0.28 | -0.25 | 0.36 | -0.19 | -0.32 | 0.42 | -0.21 | 0.14 | -0.12 | 0.02 | -0.22 | -0.24 | -0.12 | -0.51 | 0.15 | 1.00 | | | | | |
| **ImmGran%** | 0.12 | 0.32 | -0.60 | 0.58 | -0.08 | -0.06 | 0.50 | -0.09 | 0.01 | -0.09 | 0.22 | 0.04 | -0.10 | 0.16 | -0.20 | -0.18 | 1.00 | | | | |
| **Urine WBC** | -0.01 | 0.35 | 0.80 | -0.16 | 0.14 | -0.08 | -0.13 | 0.29 | 0.01 | -0.19 | -0.06 | 0.02 | -0.10 | 0.09 | -0.16 | -0.01 | -0.14 | 1.00 | | | |
| **Glucose** | -0.07 | -0.12 | -0.42 | -0.01 | 0.08 | -0.18 | -0.02 | 0.18 | -0.03 | 0.20 | -0.03 | 0.10 | 0.10 | 0.30 | -0.27 | -0.26 | -0.01 | -0.13 | 1.00 | | |
| **Lactic Acid** | 0.31 | -0.04 | 0.36 | -0.05 | 0.08 | -0.09 | -0.05 | -0.01 | -0.01 | -0.05 | 0.15 | -0.11 | 0.01 | 0.05 | 0.03 | -0.12 | -0.03 | 0.01 | 0.12 | 1.00 | |
| **CRP** | 0.61 | 0.27 | -1.00 | 0.51 | 0.00 | -0.05 | 0.42 | -0.36 | 0.71 | 0.04 | 0.63 | -0.01 | -0.03 | -0.11 | -0.21 | -0.10 | 0.59 | 0.13 | 0.26 | 0.37 | 1.00 |

**Fig 3. Intercorrelation matrix of clinical and experimental parameters.** The relationship between clinical laboratory values, such as the WBC Count, and experimental biomarkers, such as DEFA1 RNA levels, was evaluated across all 72 patients in the cohort. It is possible that some parameters, such as CRP, could have smaller n due to that test not being ordered on all patients. Color scale highlights negative (green) to positive (red) correlations. Correlations above ±0.35 are boxed to highlight possible associations.

IL8RB/CXCR2 scores compared to 40% (6/15) of patients with qSOFA scores = 0 or 1 (chi-square p < 0.05 qSOFA ≥ 2 vs qSOFA = 1 and 0).

**Defensin protein in plasma vs DEFA1 RNA.** Suspecting that rapidly decreasing DEFA1 RNA could be indicative of active translation leading to translational decay of the RNA, we analyzed the protein levels. In a subset of 15 patients and 6 controls, 29 samples (some at 0 and 3 hr) had both DEFA1 RNA by ddPCR and defensin-α1 protein in plasma by ELISA. There was a broad range of absolute values (DEFA1 RNA 0.69−87% of ACTB) and defensin-α1 protein in plasma (44−451 ng/ml), but there was only a modestly negative correlation between their levels (R = −0.27) suggesting that lower DEFA1 RNA does not directly translate to higher defensin protein levels in plasma (Fig 5). A possible explanation is that defensin-α1 protein could remain cell-associated, and especially tightly incorporated into peri-neutrophil extracellular traps (NETs), as has been observed for related proteins such as neutrophil elastase and histone antimicrobial peptides [27].

**Blood burden of bacteria versus RNA biomarkers and sepsis.** The blood levels of bacterial and fungal pathogens was quantitated by DNA purification from the Tempus nucleic acid preserved blood, followed by ddPCR using primers and probes optimized to minimize cross-over reactivity with human sequences. Separate primer/probe pairs detected Gram (+), Gram (-) 16S ribosomal bacterial sequences, and 28S fungal sequences. Blood DNA was available and successfully purified and amplified on a subset of 59 of 72 cases (all 0 hr). As shown in Fig 6 (Left panel), the pathogen DNA burden in blood showed step-wise increases from no infection, sepsis, though severe sepsis/septic shock, with Gram (-) and total reads significantly elevated above No Infection cases.

The patients were re-grouped according to the results of blood culture status, as shown in Fig 6 (Right panel). As expected, the 16S reads were highest in patients with positive blood cultures (n = 7), relative to culture-negative patients (p = ns, n = 43) or patients from whom blood cultures were not drawn (p < 0.05, n = 10). Notably, the blood culture-negative patients still had Gram (-) 16S counts more than 2-fold higher than patients not cultured, and not significantly different from blood culture-positive patients.

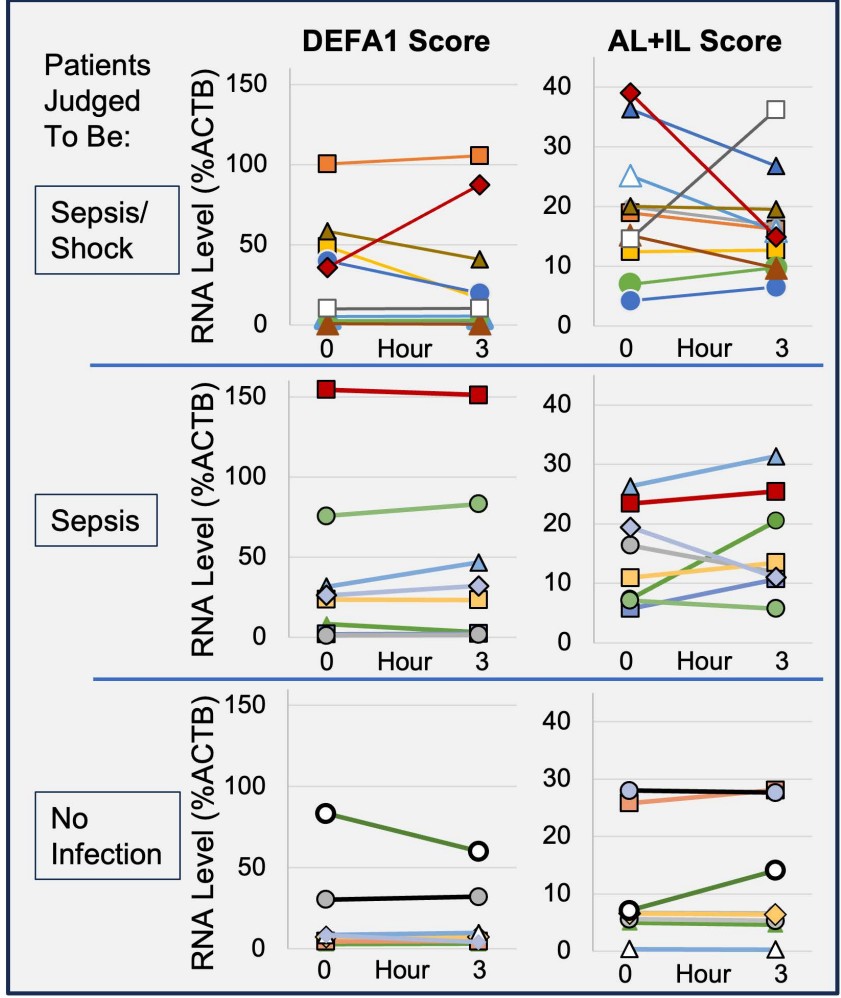

**Fig 4. Temporal changes in host immune biomarkers associated with more severe forms of sepsis.** A subset of patients were enrolled in which 0 hr and 3 hr samples were obtained. Each line reflects the same patient at two time points for bacterial response markers: DEFA1 RNA, (left panels), and ALPL+IL8RB/CXCR2 (right panels). The panels show patients judged to have more severe forms of sepsis (Sepsis/Shock, n = 11), Sepsis (n = 8), and No infection (n = 8).

**Relationship between blood pathogen levels and host RNA biomarkers.** It was then determined whether the levels of bacterial reads in blood are related to the levels of the neutrophil activation RNA biomarkers. This was evaluated in two ways. First, the correlation between microbial pathogen DNA and the neutrophil RNA biomarkers was examined. As shown in Fig 7 (Left Panel), there is a modestly positive correlation overall of R = 0.35 between the 2 parameters.

Specific cases, highlighted in Fig 7, demonstrate examples of cases in which the two parameters are highly correlated or highly disconnected. One more severe sepsis case in particular (S37), demonstrates highly elevated levels of both bacterial Gram (+) and Gram (-) DNA, as well as host RNA biomarkers (DEFA% = 100%, ALPL and IL8RB/CXCR2 = 19%). However, two other cases readily demonstrate the potential disconnects: a more severe urosepsis case with obstructive stones (S57), demonstrated even higher Gram (+) and similar Gram (-) levels, but had markedly lower bacterial RNA biomarker levels (DEFA = 4%) and equivalent ALPL and IL8RB/CXCR2 levels (ALPL + IL8RB/CXCR2 = 19%) that were just above the abnormal threshold of 18%. Conversely, a non-severe sepsis pneumonia case (S44), showed very high DEFA1

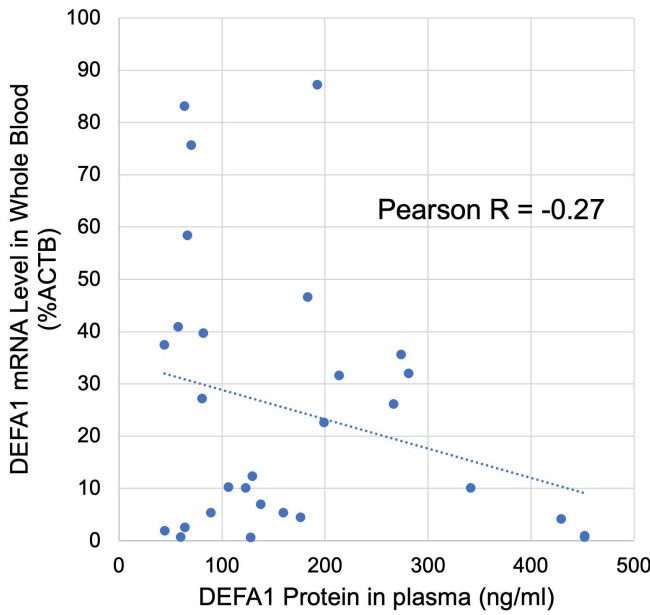

**Fig 5. DEFA1 RNA versus defensin-α1 protein levels.** In a subset of patients, it was possible to determine both the DEFA1 RNA levels by ddPCR (Y axis) and defensin-α1 protein levels by ELISA (X axis). In 29 samples, from 15 patients and 6 controls, the best linear fit is shown along with the Pearson's R correlation.

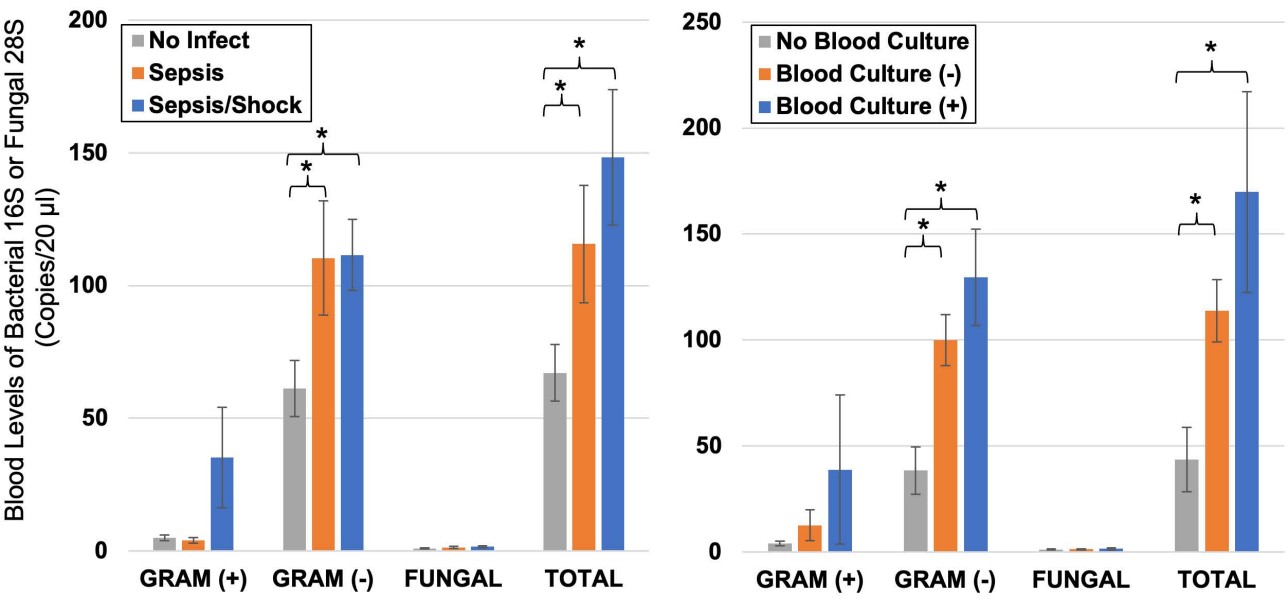

**Fig 6. Blood pathogen levels by ribosomal ddPCR versus clinical infection grouping.** Blood samples in RNA preservative were extracted for DNA, quantitated, and 200 ng was assayed by ddPCR for bacterial 16S specific for Gram (+) or Gram (-), or 28S ribosomal DNA for fungus. The read counts per pathogen type is averaged across the specific groups, with error bars reflecting s.e.m. In the left panel, the patients are grouped by the adjudicated presence of No Infection (n = 20), Sepsis (n = 20), or more severe Sepsis/Septic shock (n = 19). In the right panel, the patients are grouped by those with positive blood cultures (n = 7), versus negative blood cultures (n = 43) or no blood culture was performed (n = 10). * indicates $p < 0.05$ for comparison in brackets.

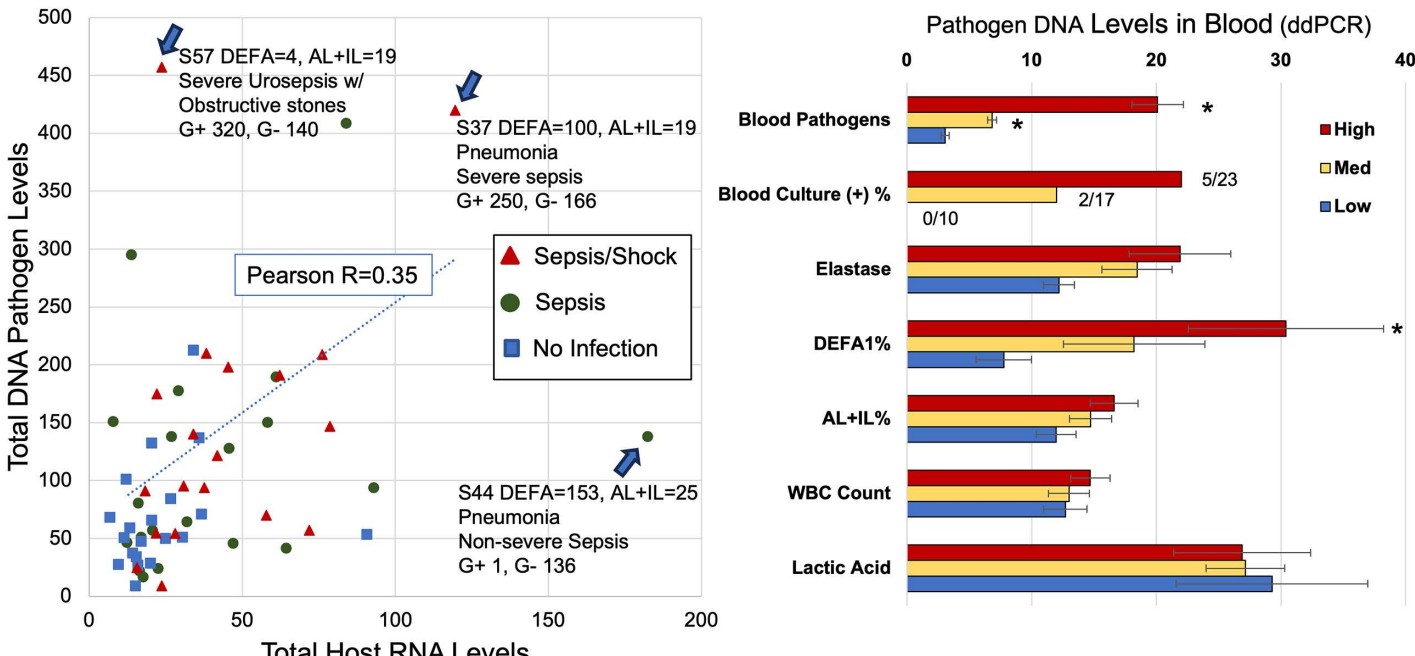

**Fig 7. Microbial pathogen levels in blood versus host RNA biomarkers.** Whole blood in a nucleic acid preservative was extracted for both RNA and DNA. The RNA was quantitated by ddPCR for the host immune RNA biomarkers, and the DNA was quantitated for Gram (+) and Gram (-) 16S ribosomal reads and fungal 28S ribosomal reads using ddPCR. In the Left Panel, each patient is plotted for total DNA levels of pathogens on the Y axis, and the total host biomarker RNA levels on the X axis. Symbols reflect the assignment of the patient to adjudicated diagnosis. In the Right Panel, the patients are stratified by the presence of High (>100 reads/20 µl), Medium (>50 but <100 reads/20 µl), or Low (<50 reads/20 µl) levels of Total Blood Pathogen reads (bacterial and fungal). Average microbial pathogens (±s.e.m) are shown in the top bars divided by 10 for graphic fit. Levels of elastase activity (Vmax), DEFA1%, ALPL+IL8RB%, WBC count (k/ul), and lactic acid (X 10) are shown as grouped by pathogen burden. * indicates $p < 0.05$.

levels (153%), and elevated ALPL and IL8RB/CXCR2 levels (25%), but had strikingly lower bacterial 16S reads (136), that were however, still above levels observed on average in the No Infection cases (mean = 61). Clearly however, there is an abundance of severe sepsis/septic shock cases with microbial DNA levels at or below those of patients judged to be not infected.

Examined differently, the general relationship between the bacterial load and RNA biomarkers is evident. As shown in Fig 7 (Right Panel), the blood bacterial DNA levels were used to divide the patients into Low (<50, n = 16), Medium (50–100, n = 20), and High burden (>100, n = 23) patients (note levels/10 in figure for graphic purposes). As expected, in the subset of cases where both the bacterial burden and blood cultures were obtained (n = 50), there was a stepwise increase in positive blood cultures from 0/10 in the low burden group to 5/23 in the high burden group. While encouraging, it highlights that of the 23 patients with the highest 16S burden in blood, only 5 showed positive blood cultures. The neutrophil elastase activity also tracked the pathogen load, but did not reach statistical significance. Further, the DEFA1% is significantly higher in the High (30.4%) vs Low (7.8%) burden group, with the Medium burden showing an intermediate level (18.2%). The ALPL and IL8RB/CXCR2 scores trended similarly, but were not statistically significant. The WBC count and lactic acid levels showed no notable variation across the bacterial burden groups.

## Discussion

Due to its rapid and often fatal course, sepsis remains a clinical challenge and active research topic. Contrary to our initial literature-based hypothesis, infected patients that developed more severe sepsis did not show levels of neutrophil

activation-related RNA biomarkers significantly different from subjects that did not develop more severe forms of sepsis. Also contrary to our expectations, we observed rapid change in these RNA biomarkers between zero and three hours among subjects that went on to develop more severe sepsis. One possible explanation for increases is that this change in measurable RNA coincided with activation of the neutrophil NETosis program [28]. Declining neutrophil activation RNA levels in some subjects may be associated with more advanced stage NETosis in which neutrophil transcription is outweighed by translational decay of RNAs.

There are two types of human defensins: α-defensins produced mostly by neutrophils and some cells in the gut; β-defensins are produced by various types of epithelial cells and leukocytes [29,30]. Both types defend against microbial infections including bacterial, fungal, viral and protozoal [31]. β-defensins have previously been reported to have reduced inducibility in leukocytes from severely septic patients [32], and β-defensin expression in whole blood from ICU patients with sepsis was found to be down-regulated [39].

Because of their direct mechanistic involvement in neutrophil function and sepsis progression, considerable work has focused on neutrophil α-defensins as biomarkers. Defensins are critical to fighting infections, but like many aspects of innate immunity can also have deleterious effects on endothelial tissues (reviewed in [33]). Studies have reported higher sepsis mortality among mice with higher copy numbers of alpha-defensin genes, and among mice treated with human alpha defensin in a mouse sepsis model [45].

Plasma levels of α-defensin proteins have been reported to be elevated in severe sepsis, and positively correlated with the SOFA scores [34]. Elevated circulating levels of defensin-α1 protein were significantly higher in COVID19 patients versus controls and associated with adverse outcomes [35]. Analysis of hemadsorption filters from septic patients suggests that defensins are major factors that could contribute to secondary endothelial damage that accelerates the syndrome [36,37]. We did not observe similar correlation between outcomes and raw defensin-α1 protein levels in our study cohort; however, changes in the levels of DEFA1 RNA over a 3-hour period was associated with clinical decompensation and emergence of more severe sepsis/septic shock. This observation in ED patients between 0 and 3 hours may be meaningful in an immunologic sense, and useful diagnostically.

Prior time series analysis of septic patients identified three genes in the elastase family (ELANE, CTSG, PRTN3) and DEFA4 as candidate biomarkers for sepsis [38], while other studies likewise implicated NETosis genes including ALPL [39]. IL8, via IL8RB/CXCR2, can trigger NETosis of neutrophils [40]. We suspect that the DEFA1/3 locus would have also been identified in these studies, but it is prone to filtering due to its multi-copy number variants. A similar approach to the present study was used to build a seven gene expression model, including DEFA4, that predicted sepsis severity with reasonable accuracy (ROC = 0.88) [41].

The present results suggest that prior biomarker studies of sepsis would have been complicated by the quite variable time frames of sampling, whereby even a 3-hour difference in sampling could change the RNA expression level by 50%. Thus, we propose a key factor in understanding the disparate results with defensin RNA levels in sepsis is the timing of blood sample in relation to the course of the disease. The present samples were captured in the ED as close as is practically possible to the time of patient presentation, while most studies have used patients with established sepsis that may have been ongoing for days. A second major factor is the difference between the RNA levels and protein levels, which is emphasized by the observation that RNA and protein levels for defensin-α1 had a slightly negative correlation. We propose that the negative trajectory of DEFA1 RNA is part of the key pathological process of NETosis, whereby the defensins, and other proteins, are translated and secreted into a physical NET that adheres either to other circulating cells or to endothelium.

The present studies are limited by a modest sample size in the trajectory analysis where only 27 patients were available with multiple time points. The enrollment criteria for sepsis was somewhat modified by the 'sepsis alert' used in the Cerner EHR, and post-hoc adjudication is subjective, although this was minimized by engaging multiple judges.

## Conclusions

○ A change in RNA markers of neutrophil activation among septic ED patients between 0 and 3 hours is associated with progression to more severe forms of sepsis.

○ Patients that developed more severe forms of sepsis did not have higher neutrophil activation RNA levels than those with sepsis.

○ Patients with more severe sepsis had similar burdens of bacteria in blood compared to those with sepsis.

## Supporting information

**S1 Table. Full demographics and lab values of patients and controls.**
(DOCX)

**S2 Table. Patient-level raw data for gene expression, diagnosis, and lab values.**
(XLSX)

**S1 Fig. Temporal changes in RNA biomarkers grouped by Sepsis-3 Criteria.**
(DOCX)

## Acknowledgments

The authors are very grateful to the patients and their families who kindly agreed to participate in this research study despite their difficult health situations. The authors are very grateful for the support of The Kasimov Family and The St. Laurent Family, whose generosity made this work possible.

## Author contributions

**Conceptualization:** John E. Lafleur, Richard Wargowsky, Kevin Jaatinen, Mary Pasquale, David Yamane, Andrew Meltzer, Timothy A. McCaffrey.

**Data curation:** John E. Lafleur, Eduard Shaykhinurov, John Perkins, Richard Wargowsky, Kevin Jaatinen, Mary Pasquale, Grace Holloway, David Yamane, Akhil Patel, Daniel King, Andrew Meltzer, Ryan Heidish, Soroush Shahamatdar, Aditya Loganathan, Tarun Loganathan, Taylor Bolden, Michael Zane Hayden, Aditya Maddali, Jennifer Goldman, Zachary Falk, Tisha Jepson, Avery League, Timothy A. McCaffrey.

**Formal analysis:** John E. Lafleur, Kevin Jaatinen, Akhil Patel, Daniel King, Michael Zane Hayden, Jennifer Goldman, Avery League, Timothy A. McCaffrey.

**Funding acquisition:** Timothy A. McCaffrey.

**Investigation:** John E. Lafleur, Eduard Shaykhinurov, John Perkins, Richard Wargowsky, Kevin Jaatinen, Mary Pasquale, Grace Holloway, David Yamane, Akhil Patel, Daniel King, Andrew Meltzer, Ryan Heidish, Soroush Shahamatdar, Aditya Loganathan, Tarun Loganathan, Taylor Bolden, Michael Zane Hayden, Aditya Maddali, Jennifer Goldman, Zachary Falk, Timothy A. McCaffrey.

**Methodology:** John E. Lafleur, Eduard Shaykhinurov, John Perkins, Richard Wargowsky, Kevin Jaatinen, Mary Pasquale, David Yamane, Akhil Patel, Daniel King, Andrew Meltzer, Ryan Heidish, Soroush Shahamatdar, Aditya Loganathan, Tarun Loganathan, Taylor Bolden, Michael Zane Hayden, Jennifer Goldman, Zachary Falk, Timothy A. McCaffrey.

**Project administration:** John E. Lafleur, Eduard Shaykhinurov, John Perkins, Richard Wargowsky, Mary Pasquale, David Yamane, Akhil Patel, Daniel King, Andrew Meltzer, Ryan Heidish, Soroush Shahamatdar, Aditya Loganathan, Tarun Loganathan, Tisha Jepson, Timothy A. McCaffrey.

**Supervision:** John E. Lafleur, David Yamane, Akhil Patel, Andrew Meltzer, Ryan Heidish, Soroush Shahamatdar, Aditya Loganathan, Tarun Loganathan, Tisha Jepson, Timothy A. McCaffrey.

**Validation:** Richard Wargowsky, Kevin Jaatinen, Grace Holloway, Jennifer Goldman, Zachary Falk, Avery League, Timothy A. McCaffrey.

**Visualization:** John Perkins, Richard Wargowsky, Kevin Jaatinen, Grace Holloway, Michael Zane Hayden, Tisha Jepson, Avery League, Timothy A. McCaffrey.

**Writing – original draft:** John E. Lafleur, Timothy A. McCaffrey.

**Writing – review & editing:** John E. Lafleur, Avery League.

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
