## [Decision Letter · Decision Letter 0]

25 Jun 2025

Dear Dr. McCaffrey,

Thank you for submitting your manuscript to PLOS ONE. After careful consideration, we feel that it has merit but does not fully meet PLOS ONE’s publication criteria as it currently stands. Therefore, we invite you to submit a revised version of the manuscript that addresses the points raised during the review process.

We look forward to receiving your revised manuscript.

Kind regards,

Claudine Irles, Ph.D.

Academic Editor

PLOS ONE

Journal Requirements:

“The Ulvi and Reykhan Kasimov Family

The St. Laurent Institute

NIH S10 OD021622”

“The authors are very grateful to the patients and their families who kindly agreed to participate in this research study despite their difficult health situations.  The authors are grateful to The Ulvi and Reykhan Kasimov Family and The St. Laurent Institute for generous financial support that made these studies possible.  Other support was obtained from the NIH S10 OD021622 to TM.”

“The Ulvi and Reykhan Kasimov Family

The St. Laurent Institute

NIH S10 OD021622”

Reviewers' comments:

Reviewer's Responses to Questions

**Comments to the Author**

1. Is the manuscript technically sound, and do the data support the conclusions?

Reviewer #1: Yes

2. Has the statistical analysis been performed appropriately and rigorously?

Reviewer #1: Yes

3. Have the authors made all data underlying the findings in their manuscript fully available?

Reviewer #1: Yes

4. Is the manuscript presented in an intelligible fashion and written in standard English?

Reviewer #1: Yes

Reviewer #1: Introduction:

The authors appropriately delineate the limitations of the prior definitions of sepsis that led to the current definition often referred to as ‘Sepsis-3.’ As noted, a foundational flaw in the prior definitions of sepsis is their requirement for SIRS criteria, which are neither sensitive nor specific. Further, while it is true that the SOFA score was developed to quantify and characterize critical illness, it is also, to various degrees, included in the more contemporary definition of sepsis. In any case, this discussion of the various definitions of sepsis – and their limitations – becomes important because, in the results and discussion, it appears that the authors have reverted to the prior definitions.

The discussion regarding the treatment of sepsis, while true, is largely unnecessary and impertinent to the current discussion, which is one of diagnosis (not treatment).

The results are ‘teased’ in the introduction, which is somewhat unusual and should be reserved for the results and discussion sections.

Methods

The authors point to the criteria used in the sepsis alert; however, these criteria are not clear. Please include the referenced figure (i.e., are these the critera noted as ‘enrollment criteria’ in figure 1?

In using these criteria, it should be noted that these are not SIRS criteria, but they are a modified version of SIRS. For example, the Temp > 38.3 is higher than traditional SIRS criteria, as are both the Respiratory and Heart Rates. In this way, this modification of SIRS criteria will be less sensitive than previously noted; it is unclear the impact that the glucose will have. In essence, this will result in an included population that is somewhat different than any current definition of sepsis, and may impact the generalizability of the results.

How was the subset of patients and controls selected for measurement of the defensin-�1 level selected? Was this selection random, or was it based on specific criteria, or perhaps through more practical reasons (availability of sample for testing).

With regards to controls … in many cases, understanding the control group becomes one of the most integral components to interpreting a study. There is mention that a normal control group was derived from medical center personnel, and, presumably, another group of subjects with self-reported infections was enrolled. How were these subjects selected? Who are they? A description is important because a 25 year-old medical resident is vastly different than a 65 year old ED patient; the mechanism of selection of the control group is important to disclose.

Results

Overall, the number of groups included in the study is somewhat difficult to follow (there are 5 – control, self-reported infection, no sepsis, sepsis, septic shock). There may be a way to simplify this, and the N’s of the control groups are small enough that it is unclear that they provide any meaningful information.

The first paragraph of the results is the first mention of adjudication for sepsis and sepsis shock. Because there is no reference standard for sepsis, it is necessary to have such a process. This does raise several questions: how was the adjudication performed? Why did the authors chose to use the now antiquated terminology of ‘sepsis / severe sepsis / septic shock,’ which were abandoned in the current Sepsis-3 definition referenced in the Introduction. Also, as previously noted, the vital sign criteria are modified from the original SIRS criteria such that these patients may or may not meet any particular definition of sepsis. An explanation for these criteria and a summary of the adjudication process should be included in the Methods.

“Average lactate levels were artificially high …” This statement is incorrect; there is nothing ‘artificial’ about the lactic acidosis associated with such physiologic derangements as alcohol withdrawal and seizure.

Table 1: This table needs a legend. Are these means, medians? It should also be reformatted to read easier. Blood culture (+) – is this percent? Presumably so – but is this percent of cultures drawn or percent of patients? We could presume that all patients had blood cultures drawn, but this presumption may not be accurate. Where appropriate, units should be included. I note that, in race, the %white is on average, about 33%. Since this represents a minority of the study population, it would be helpful to have more information on the ethnic and racial distribution. Also, the number of patients in each group is important, given than many of the patients are unlikely to have had specific values tested (CRP, lactic acid, etc), given the data were derived from standard care.

The control group is not included in any description. They should be included in Table 1, if not elsewhere. It is possible that they are in an original format, but, for the PDF available for review, I do not see a column for controls.

Again, a more detailed explanation or description of the adjudication process would be appropriate. It is difficult to interpret inter-observer variability for infection and sepsis without understanding the process by which the determination was made, especially since this definition of sepsis is not consistent with any of the prior definitions that have been published.

The term “simple sepsis” is used. Colloquially, this is often the term we use for ‘SIRS with a source.’ This is not a technical term, and it should be defined or, better yet, abandoned. There is some mention of ‘non-ED controls with self-reported infections’ in the methods; however, there is no detail in how these controls were selected. More clarity is necessary as mentioned in the above comments about the methods.

For the investigation in to termporal change – did this include all individuals? This should be included in the Methods; the first mention of this should not be in Results. In examining Figure 4, it appears that the serial measurement was only performed in a subset of patients, and this is confirmed in the legend, so a description of why and how this occurred is helpful.

The statement “suggesting that blood culture misses a significant percentage of bacteremic cases” is interpretive and should be moved to the Discussion.

In Figure 7, specific cases are highlighted. How and why were these cases chosen? What is different about them as compared to the ones that were not included? Some mechanism to disclose this information would be appreciated.

That said, there are considerable limitations given the quite restrictive inclusion criteria that were used. I would certainly welcome the opportunity to review this manuscript again after revision, because I do think, in light of the current advancement in diagnostics – including expression assays – such knowledge and understanding is helpful.

In summary, though I find the concept and results intriguing and certainly worthy of publication, in light of the contemporary definition of sepsis, I find the results, as presented, difficult to interpret. Perhaps the results would be different if it were presented as ‘no infection,’ ‘infection,’ and ‘sepsis’ by the Sepsis-3 definitions. Unfortunately, it is difficult to assess given the small sample size and heterogeneity seen in the study population vs the control group, and without great understanding of the adjudication process. In any case, the findings suggest that temporal changes in gene expression are important; thus, reformatting of the manuscript may make these results more easily interpreted in the context of our current definitions. Whether or not this is clinically meaningful is in question, but this small (though rigorous) study is meant to be hypothesis generating.

Thoughout the manuscript, then N’s are very different in each group. For example, in Figure 6, the N’s are 20 / 20 / 19 for No Infection / Severe Sepsis / Septic Shock while in Table 1, the N’s are 24 / 26 / 22. An explanation or description of the missing patients would be helpful.

**Do you want your identity to be public for this peer review?** For information about this choice, including consent withdrawal, please see our Privacy Policy

Reviewer #1: No

---

## [Author Response · Author response to Decision Letter 1]

1 Aug 2025

Response to Critique: LaFleur et al. Blood RNA biomarkers and a point-of-care elastase assay for detecting host immune activation in suspected sepsis: Trajectory matters.

Reviewer #1:

We are very grateful for your thoughtful review, which is constructive, detailed, and accurate. We believe that all of the concerns are addressed to the best of our abilities, and that the manuscript is improved because of the changes. We include a ‘track changes’ version so that it is easy to identify the extensive revisions, including new data analysis, that were made to address these concerns.

Introduction:

Concern 1: The discussion regarding the treatment of sepsis, while true, is largely unnecessary and impertinent to the current discussion, which is one of diagnosis (not treatment).

Response: We have shortened this section to discuss only how diagnostic uncertainty drives needless treatments.

Concern 2: The results are ‘teased’ in the introduction, which is somewhat unusual and should be reserved for the results and discussion sections.

Response: We have also shortened this section to simply frame the question, without mentioning the outcomes.

Methods:

Concern 3: The authors point to the criteria used in the sepsis alert; however, these criteria are not clear. Please include the referenced figure (i.e., are these the criteria noted as ‘enrollment criteria’ in figure 1?

In using these criteria, it should be noted that these are not SIRS criteria, but they are a modified version of SIRS. For example, the Temp > 38.3 is higher than traditional SIRS criteria, as are both the Respiratory and Heart Rates. In this way, this modification of SIRS criteria will be less sensitive than previously noted; it is unclear the impact that the glucose will have. In essence, this will result in an included population that is somewhat different than any current definition of sepsis, and may impact the generalizability of the results.

Response: These are valid points. Yes, the values in Figure 1 are the criteria used for the Cerner ‘sepsis alert’ that identified subjects for screening and consent. The Cerner alert is based on the St. John’s sepsis alert, which is now cited for clarity and fully detailed in Methods. While it is slightly different than the SIRS criteria, it is a standardized system that many ED physicians are familiar with. We corrected all ‘SIRS criteria’ to ‘sepsis alert’, which is clearly defined as a modified SIRS criteria in the methods. Further, toward a similar point discussed later, we went back to the medical records and computed qSOFA scores used in the SEPSIS-3 criteria (see below).

Concern 4: How was the subset of patients and controls selected for measurement of the defensin-�1 level selected? Was this selection random, or was it based on specific criteria, or perhaps through more practical reasons (availability of sample for testing).

Response: Yes, the selection of samples for defensin-a1 protein level was made purely on the basis of having plasma available for testing. This is now noted in Methods.

Concern 5: With regards to controls … in many cases, understanding the control group becomes one of the most integral components to interpreting a study. There is mention that a normal control group was derived from medical center personnel, and, presumably, another group of subjects with self-reported infections was enrolled. How were these subjects selected? Who are they? A description is important because a 25 year-old medical resident is vastly different than a 65 year old ED patient; the mechanism of selection of the control group is important to disclose. The control group is not included in any description. They should be included in Table 1, if not elsewhere. It is possible that they are in an original format, but, for the PDF available for review, I do not see a column for controls. There is some mention of ‘non-ED controls with self-reported infections’ in the methods; however, there is no detail in how these controls were selected. More clarity is necessary as mentioned in the above comments about the methods.

Response: The control group, including those with self-reported infections, are now more fully described in the text. We prepared summary statistics of age, sex, and race, and this is included as Supplementary Table 1. It would not be practical to include the controls in Table 1 because there are no lab values available on these non-patient controls. However, we have repeated the other groups in the supplementary table so that they can easily be compared. In short, the controls are younger and more likely to be White, but their main purpose is simply to provide a reference for non-ED patients, with or without self-reported infection, and they fulfill that need. We extensively revised Table 1 to include full titles for the variables, with additional details in the legend.

Results:

Concern 6: Overall, the number of groups included in the study is somewhat difficult to follow (there are 5 – control, self-reported infection, no sepsis, sepsis, septic shock). There may be a way to simplify this, and the N’s of the control groups are small enough that it is unclear that they provide any meaningful information.

Response: There are 16 controls without any self-reported infection, and this adds a valuable frame of reference for the 8 sub-clinical infections, and the total of 72 ED patients. The n is sufficient to detect statistically significant differences in the RNA biomarker levels (Figure 2). Based on the new Sepsis-3 analysis, we provide new groupings that are simpler, and more amenable to the terminology moving forward.

Concern 7: The first paragraph of the results is the first mention of adjudication for sepsis and sepsis shock. Because there is no reference standard for sepsis, it is necessary to have such a process. This does raise several questions: how was the adjudication performed? Why did the authors choose to use the now antiquated terminology of ‘sepsis / severe sepsis / septic shock,’ which were abandoned in the current Sepsis-3 definition referenced in the Introduction. Also, as previously noted, the vital sign criteria are modified from the original SIRS criteria such that these patients may or may not meet any particular definition of sepsis. An explanation for these criteria and a summary of the adjudication process should be included in the Methods. […] Again, a more detailed explanation or description of the adjudication process would be appropriate. It is difficult to interpret inter-observer variability for infection and sepsis without understanding the process by which the determination was made, especially since this definition of sepsis is not consistent with any of the prior definitions that have been published.

Response: True, we used the prior Sepsis-2 terms because those are still terms relevant to billing in many US centers, and are still very familiar to most practicing physicians. We agree that Sepsis-3 terms will be more relevant moving forward, and so we reanalyzed the medical records to build the Sepsis-3 qSOFA scores and we present similar results using this new analysis.

The adjudication process is now explained in greater detail, and it is supplemented by the new analysis using the Sepsis-3 criteria to group the patients. We think that with the clarification of the Cerner Sepsis alert, the clarified adjudication process, and the new qSOFA analysis that readers will be able to understand the patient groups adequately. We show the results of the qSOFA analysis as new panels in Figure 2, and as a new Supplementary Figure 1 for the trajectories. Further, to clarify the group names in this analysis, we changed “Severe Sepsis” to “Sepsis/Shock” to indicate that this group is more severe sepsis and includes patients with organ failure signs and vasopressors in some cases.

Concern 8: “Average lactate levels were artificially high …” This statement is incorrect; there is nothing ‘artificial’ about the lactic acidosis associated with such physiologic derangements as alcohol withdrawal and seizure.

Response: Apologies, we meant that the elevated lactate was not related to infection. We have corrected the wording to “higher than expected”.

Concern 9: Table 1: This table needs a legend. Are these means, medians? It should also be reformatted to read easier. Blood culture (+) – is this percent? Presumably so – but is this percent of cultures drawn or percent of patients? We could presume that all patients had blood cultures drawn, but this presumption may not be accurate. Where appropriate, units should be included. I note that, in race, the %white is on average, about 33%. Since this represents a minority of the study population, it would be helpful to have more information on the ethnic and racial distribution. Also, the number of patients in each group is important, given than many of the patients are unlikely to have had specific values tested (CRP, lactic acid, etc), given the data were derived from standard care.

Response: Table 1 has been completely revised with the variables spelled out and a legend that clearly denotes that they are means, sems, and how the statistics were conducted.

Concern 10: The term “simple sepsis” is used. Colloquially, this is often the term we use for ‘SIRS with a source.’ This is not a technical term, and it should be defined or, better yet, abandoned.

Response: We removed all instances of ‘simple sepsis’.

Concern 11: For the investigation into temporal change – did this include all individuals? This should be included in the Methods; the first mention of this should not be in Results. In examining Figure 4, it appears that the serial measurement was only performed in a subset of patients, and this is confirmed in the legend, so a description of why and how this occurred is helpful.

Response: The 0 and 3 hour measures were only available on a subset of patients (n=27), largely because the patient declined the second draw or was otherwise unavailable due to transfer or discharge. We have clarified this subset as patients in which their clinical course and consent allowed a 2nd draw.

Concern 12: The statement “suggesting that blood culture misses a significant percentage of bacteremic cases” is interpretive and should be moved to the Discussion.

Response: Agreed. This is now deleted.

Concern 13: In Figure 7, specific cases are highlighted. How and why were these cases chosen? What is different about them as compared to the ones that were not included? Some mechanism to disclose this information would be appreciated.

Response: Agreed. We noted that these examples are chosen because they represent ‘non-synonymous’ cases of pathogen load and RNA biomarker levels. We explain more clearly that they are cases where pathogen levels and the RNA biomarkers are “highly correlated or highly disconnected”.

Concern 14: Throughout the manuscript, then N’s are very different in each group. For example, in Figure 6, the N’s are 20 / 20 / 19 for No Infection / Severe Sepsis / Septic Shock while in Table 1, the N’s are 24 / 26 / 22. An explanation or description of the missing patients would be helpful.

Response: True. The changing N’s are due to the specific criteria involved in each of the figures and reflect our best efforts to be accurate about the size of the groups in each comparison. In Figure 6, for example, we did not always have excess blood for the isolation of whole blood DNA, and thus the N is smaller. We have added text to clarify this.

Final Issues:

That said, there are considerable limitations given the quite restrictive inclusion criteria that were used. I would certainly welcome the opportunity to review this manuscript again after revision, because I do think, in light of the current advancement in diagnostics – including expression assays – such knowledge and understanding is helpful.

In summary, though I find the concept and results intriguing and certainly worthy of publication, in light of the contemporary definition of sepsis, I find the results, as presented, difficult to interpret. Perhaps the results would be different if it were presented as ‘no infection,’ ‘infection,’ and ‘sepsis’ by the Sepsis-3 definitions. Unfortunately, it is difficult to assess given the small sample size and heterogeneity seen in the study population vs the control group, and without great understanding of the adjudication process. In any case, the findings suggest that temporal changes in gene expression are important; thus, reformatting of the manuscript may make these results more easily interpreted in the context of our current definitions. Whether or not this is clinically meaningful is in question, but this small (though rigorous) study is meant to be hypothesis generating.

---

## [Decision Letter · Decision Letter 1]

6 Nov 2025

Dear Dr. McCaffrey,

Thank you for submitting your manuscript to PLOS ONE. After careful consideration, we feel that it has merit but does not fully meet PLOS ONE’s publication criteria as it currently stands. Therefore, we invite you to submit a revised version of the manuscript that addresses the points raised during the review process.

We look forward to receiving your revised manuscript.

Kind regards,

Tomasz W. Kaminski

Academic Editor

PLOS ONE

Journal Requirements:

Additional Editor Comments:

Dear Authors,

Thank you for your thorough and thoughtful revision. The revised version of your manuscript shows substantial improvement and clearly addresses the major reviewer concerns from the previous round. I appreciate the extensive effort invested in refining the analyses and clarifying the methods and results.

At this stage, I have only a few minor points that should be addressed before the manuscript can proceed to acceptance. These are primarily editorial and clarificatory in nature:

-State statistical approach in more detail.

Briefly describe normality testing, justify the use of parametric or non-parametric tests,

-Ensure consistent figure and table formatting.

Verify that figure numbering matches the text; add missing units to all axes and continuous variables in tables.

- Add or confirm data availability information.

Please provide a DOI or repository link for the underlying data, as required by journal policy.

-Clarify the “No Sepsis” group definition.

Please explicitly define this cohort as sepsis-alert positive but non-infectious inflammatory controls to avoid confusion regarding elevated CRP and inflammatory markers.

- Expand the limitations paragraph slightly.

Explicitly acknowledge the small trajectory subset, modified enrollment criteria, and adjudication variability.

- Proofread for minor typographical and formatting inconsistencies.

(e.g., spacing, capitalization in figure labels, consistent terminology such as “flu” vs “influenza.”)

All the best.

Reviewers' comments:

Reviewer's Responses to Questions

**Comments to the Author**

Reviewer #1: All comments have been addressed

2. Is the manuscript technically sound, and do the data support the conclusions?

Reviewer #1: Yes

3. Has the statistical analysis been performed appropriately and rigorously?

Reviewer #1: Yes

4. Have the authors made all data underlying the findings in their manuscript fully available?

Reviewer #1: Yes

5. Is the manuscript presented in an intelligible fashion and written in standard English?

Reviewer #1: Yes

Reviewer #1: My concerns have been adequately addressed. I appreciate the author's attention and effort. My only comment is that some of the tables need appropriate formating, but this may be in the rendering of the document.

**Do you want your identity to be public for this peer review?** For information about this choice, including consent withdrawal, please see our Privacy Policy

Reviewer #1: No

---

## [Author Response · Author response to Decision Letter 2]

13 Nov 2025

Response to Editorial Concerns: (LaFleur et al. PONE-D-25-10033R1)

Overall: At this stage, I have only a few minor points that should be addressed before the manuscript can proceed to acceptance. These are primarily editorial and clarificatory in nature:

Concern: State statistical approach in more detail.

Briefly describe normality testing, justify the use of parametric or non-parametric tests.

Response: A brief “Statistical Methods” paragraph was added to address this.

Concern: Ensure consistent figure and table formatting.

Verify that figure numbering matches the text; add missing units to all axes and continuous variables in tables.

Response: We verified that figure numbering matches the text. We added several units that were missing in the figures, especially Figure 4. We could not detect any missing units in the only table, Table 1.

Concern: Add or confirm data availability information.

Please provide a DOI or repository link for the underlying data, as required by journal policy.

Response: Due to the US government shutdown, it has been impossible to get an accession at Gene Expression Omnibus (GEO), and it may take months to resolve a 40 day backlog. Because there are only 6 gene expression levels per patient, we have included the patient level raw data, in anonymous form, in Supplementary Table 2.

Concern: Clarify the “No Sepsis” group definition.

Please explicitly define this cohort as sepsis-alert positive but non-infectious inflammatory controls to avoid confusion regarding elevated CRP and inflammatory markers.

Response: Correct. We inserted a sentence to this effect in the methods.

Concern: Expand the limitations paragraph slightly.

Explicitly acknowledge the small trajectory subset, modified enrollment criteria, and adjudication variability.

Response: True, these limitations were added to the end of the discussion.

Concern: Proofread for minor typographical and formatting inconsistencies.

(e.g., spacing, capitalization in figure labels, consistent terminology such as “flu” vs “influenza.”)

Response: We corrected figure labels for consistent capitalization, and made several corrections for consistent terminology.

---

## [Editor Report · Decision Letter 2]

18 Nov 2025

Blood RNA biomarkers and a point-of-care elastase assay for detecting host immune activation in suspected sepsis: Trajectory matters.

PONE-D-25-10033R2

Dear Dr. McCaffrey,

We’re pleased to inform you that your manuscript has been judged scientifically suitable for publication and will be formally accepted for publication once it meets all outstanding technical requirements.

Kind regards,

Tomasz W. Kaminski

Academic Editor

PLOS ONE

---

## [Editor Report · Acceptance letter]

PONE-D-25-10033R2

PLOS ONE

Dear Dr. McCaffrey,

I'm pleased to inform you that your manuscript has been deemed suitable for publication in PLOS ONE. Congratulations! Your manuscript is now being handed over to our production team.

Kind regards,

on behalf of

Dr. Tomasz W. Kaminski

Academic Editor

PLOS ONE